# Biosocial correlates of adolescent's knowledge on pubertal changes in rural Bangladesh: A structural equation model

Md Injamul Haq Methun[1]*, M. Sheikh Giash Uddin[2], Md. Ismail Hossain[2], Md. Jakaria Habib[2], Ahmed Abdus Saleh Saleheen[2], Iqramul Haq[3]

1 Statistics Discipline, Tejgaon College, Dhaka, Bangladesh, 2 Department of Statistics, Jagannath University, Dhaka, Bangladesh, 3 Department of Agricultural Statistics, Sher-e-Bangla Agricultural University, Dhaka, Bangladesh

* iemthuns7@gmail.com

**Data Availability Statement:** The relevant data for this study can be found on Zenodo at https://doi.org/10.5281/zenodo.5650629.

## Abstract

### Background

The present study aimed to identify factors that are associated with puberty knowledge among school-going rural adolescents in Bangladesh.

### Methods

This cross-sectional study was conducted on 2724 school-going (grades VI-IX) adolescents who were aged between 10–24 years. The adolescents resided only in rural areas of Bangladesh. In this study, relationship between socio-demographic factors and controlling behaviour was assessed considering Bronfenbrenner's bioecological model. Considering the complex nature of Bronfenbrenner's bioecological model the structural equation model to explore factors related to the Adolescents' knowledge of pubertal changes.

### Results

The structural equation model result showed a significant association among gender, education, age, and parental limit setting on daily activities with student's knowledge on pubertal changes. peer connection, and peer regulation were associated with adolescent knowledge on puberty directly as well as through the mediator variables year of schooling, academic performance and, parental behavioural control.

### Conclusion

Adolescents Age, years of schooling, and teachers concerns are positively associated with adolescents' knowledge on puberty. Whereas, parents' and peers' controlling behaviors are negatively associated with adolescents' understanding of pubertal changes. Therefore, there is needed an effective plan to raise the attention of parents and teachers on adolescents' pubertal issues to ensure adolescents' informed pubertal period.

**Funding:** The author(s) received no specific funding for this work.

**Competing interests:** The authors have declared that no competing interests exist.

## Introduction

The onset of puberty causes physical and physiological, psychological, social, mental, cognitive, and behavioural changes [1–3]. The physical changes during puberty among girls are breast foliage, rapid height and weight rise, pubic and axillary hair development, hip widening and menstrual onset [1, 4–6]. Likewise, boys experience changes in tone, increase in shoulder width, night-time ejaculation, penis enlargement, axillary hair growth, boys' facial hair [7]. At the same time, adolescents go through a profound psychological transformation, and they experience a wide change in their self-image, attitude, and relationships. Moreover, they also experience a sudden change in social, academic, and other environmental influences [8]. Those multidimensional changes make them confused, insecure, and self-centred [9, 10]. As a result, mental disorders, such as severe depression, anxiety disorders, eating disorders, and substance use disorders, can arise during puberty [11]. More than 50 percent of female adolescents had experienced moderate to severe stress due to changes in puberty [12].

If adolescents can accept themselves and their body functioning during the adolescence period, they may experience a smooth pubertal period and establish a healthy attitude towards sex, marriage, parenthood, and family. Therefore, they need an explanation about the process of puberty that unfolds for everyone and what changes to expect. The lack of adequate knowledge and confusion about their own physical and/or sexual development (i.e., changes in growth) can expose them to depression, anxiety, conduct disorder, and self-harm [13]. Misconceptions, taboos and myths about sex, sexuality, reproduction, and contraception also hinder them from obtaining correct knowledge [14]. Thus, teenagers are getting more fragile along with their distinctive development stage.

The family is responsible for ensuring informed puberty so that adolescents can lead healthy, secure, productive, and enjoyable lives and protect themselves from reproductive health problems. However, in the socio-cultural context of Bangladesh, parents neglect their duty to pass health information to their children because of either shyness or indifference or life obligations and parents left it as a responsibility of teachers who may also overlook it [15]. In such instances, adolescents have no choice other than to go to their peers, siblings, and mass media. These sources offer incomplete and inappropriate sexual and reproductive health knowledge to the adolescents [16]. Therefore, adolescents can get inaccurate or insufficient knowledge about pubertal changes which may rise misconceptions about puberty among them [17]. With the misconceptions, adolescents can create problems for themselves and their parents by putting their physical, emotional, and social well-being at stake [15].

Both the Government of Bangladesh (GOB) and NGOs have undertaken various activities at different times to alleviate the current disparity between the need and prevalence of adequate knowledge and awareness of puberty. Those activities are implemented under the adolescent sexual and reproductive health (ASRH) program initiatives. There were 32 adolescent sexual and reproductive health awareness-raising and service delivery initiatives introduced in Bangladesh between 2005 and 2015 [18]. To make these programs successful, it is essential to identify the factors that are related to adolescent's knowledge of pubertal changes considering the interrelationship between development and the child circumambient environmental context. Several studies on adolescents' knowledge have been carried out in Bangladesh [19–23] and other low- and middle-income countries [12, 24–29]. Most of these studies ignore the interrelationship among the child and the surrounding environmental contexts such as the interrelationship among family, community, peer and school. Therefore, to make the awareness-raising program more effective, an empirical study about how family and community relates to adolescents' knowledge of pubertal changes, is to be carried out.

Therefore, in the situation of minimal evidence, this study sets out to add empirical evidence on the relationship between the environmental context of child development and adolescents' knowledge of pubertal changes in Bangladesh.

## Conceptual framework

In this study, the effect of socio-demographic factors and controlling behaviour was assessed considering Bronfenbrenner's bioecological model [30]. According to the bioecological model, environmental setting along with individual characteristics may facilitate the proximal process of adolescent functioning. In this study, the relationship between adolescents' knowledge on pubertal changes and personal characteristics such as age, sex, religion, and year of schooling was assessed.

Environmental settings and adolescent functioning are linked through social contexts and their interrelations; social context includes family, peers, school, and community [31, 32]. The contextual factors that are used in this study are—mother's education, parental behavioural control, parental limit-settings on daily activities, peer connection, peer regulation, peer psychological control, community disorganization, academic performance, and teacher concern. The proximal process of the adolescent functioning (acquiring knowledge) provides the conceptual framework (Fig 1) of this study.

## Methods

### Data

Data for this cross-sectional study was collected by the Department of Statistics of Jagannath University in March 2019 in the rural areas of Bangladesh. Data regarding socio-demographic, psychosocial, and reproductive health knowledge and status-related information were

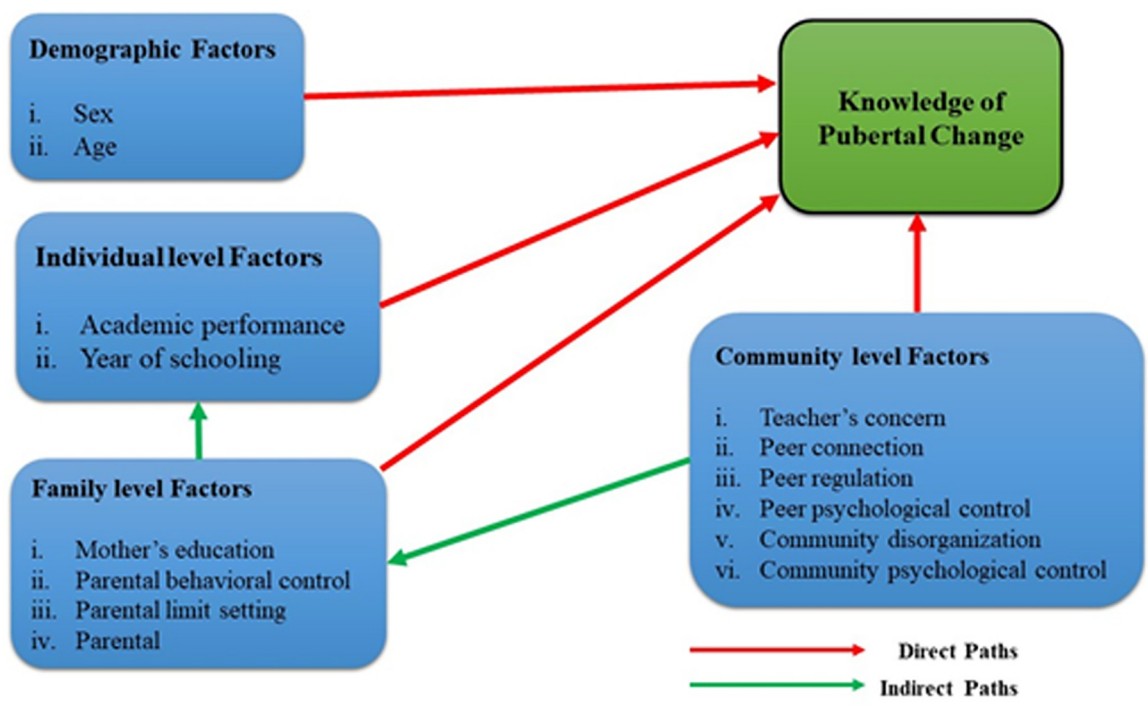

**Fig 1. Conceptual framework.**

collected from adolescents aged 10–24 years in rural areas of Bangladesh. The survey consisted of two modules of data set named a school module of data sets and household module dataset. The household module was designed to collect information from community adolescents aged 10–24 years. In the survey, 3013 adolescents were successfully interviewed in rural areas of Bangladesh, with 1504 adolescents were in the household module and 1509 adolescents were in the school module.

The survey used a three-stage stratified sampling procedure to select the adolescents. Based on the performance of reproductive health programs, at first 8 upazilas (administrative geographical unit similar to sub-district) of 8 divisions were selected randomly. At the first stage, two unions from each selected upazila were selected randomly. Then at the second stage, from each selected union one secondary school was selected randomly. Then from each secondary school, 47 male and 47 female students of class VI-IX was selected applying simple random sampling.

In the household module, one or two villages within the catchment area of the selected school or madrasa were surveyed. For the household survey, a cluster was comprised of taking 150–200 households within the catchment areas of the selected school. Before the main survey, a household listing was conducted to identify the eligible adolescents. Finally, 95 households were selected within each cluster by systematic random sampling technique. One respondent was chosen in the selected household. Listing operations were done for each and every household with household head name and demographic information. The list recorded the name of household head to select whether eligible respondents were in the household within each cluster. The resulting lists of Households with eligible study populations served as a sampling frame at the final stage.

Finally, two data modules were merged for the analysis. After deleting the missing and incomplete cases, there were 2724 cases which were used for the final analysis.

## Ethical consideration

All participants of this study were informed before data collection about the nature and purpose of the study, the procedure, and the right to withdraw their data from the study. Before data collection, verbal consent was taken from all participants, the headmasters of all secondary schools for the school module, and parents or guardians for household module. The study protocol was approved by the Jagannath University, Bangladesh institutional ethical committee (JEC) (Under the project number 37.20.0000.004.033.020.2016.7725).

## Outcome variable

The outcome variable was knowledge of pubertal changes. To assess adolescents' knowledge regarding pubertal changes, students were asked to identify the physical, mental, and behavioural changes during puberty. In the text book of classes VI, VII and VIII, eighteen physical, mental and behavioural changes were mentioned. Adolescents get score 1 for the identification of each change which was in their text book. The total number of changes one had correctly identified was the knowledge on pubertal changes score. Thus, one adolescent can score maximum 18 score and minimum 0. A similar kind of knowledge measuring approach has been used in some studies [12, 29, 33].

The latent construct approach is the most suitable approach for this type of study. However, Field et al. [34] suggested that to construct a latent variable it is essential to remove observed variables if the correlation between variables is very high or is not high-enough. In this study, there are several variables between which correlation is either very high or is not high enough for factor analysis. If those variables were removed from the study, then a considerable amount

of information has been lost on adolescents' knowledge on pubertal changes. Therefore, instead of using the latent construct of knowledge on pubertal changes, knowledge on pubertal changes was measured with constructed score [12, 28, 32].

## Control variables

**Individual factors.** Control variables were year of schooling, age, sex (male and female), religion (Islam and others), and Mother's education (no education, primary, secondary and higher).

**Contextual factors.** The contextual factors were measured by utilizing the method from Amoateng and Kalule-Sabiti [35]. They developed this method to find out the bio-social correlates of age at first sexual intercourse among students of South Africa. The method measures various social contextual factors related to family, peers, school, and community that connect the environment with adolescent functioning. For this study, the translation and adoption procedure of the method is done by following the proposed guideline of Beaton et al. [36].

*Parental behavioural control (PBC).* Parental behavioural control is a composite index. It was measured as a mean score of five questions about how much their parents are conscious about their friends, night stay, and money spending. The responses are taken into a three-point Likert scale categorized as "Does not know", "know a little", and "knows a lot".

*Parental limit settings (PLS).* To understand the parental limit-setting behaviour respondents are asked four questions about parenting actions. Which are how much did one of your parents or primary caregivers do the following things over the past 30 days: "regulate the period of time you could watch televisions", "inspect whether or not homework was done", "help to prepare home assignment, check the exam grade sheet". The responses are taken on a four-point Likert scale- 1. "Never", 2. "On rare occasion" 3. "Occasionally" 4. "Frequently". The average value of these four questions is taken as the child's parental limit setting.

*Community disorganization (CD).* The mean score of five questions related to how often they are disrupted in the community is used to assess the community's disorganization. The questions are- "How often do you see the litter or trash on the sidewalks and streets?", "How often do you see graffiti on building walls?", "How often do you see alcoholics or drug sales?", "How often do you become afraid or worried when you are walking through the empty place?", "How frequently burglary happened?". The responses range from 1 to 3. 1 stands for "Often", 2 for "Sometimes" and 3 for "Never".

*Community psychological control (CPC).* The average response of three five-point Likert scale questions is used as community psychological control. The problems are related to is your neighbours always watch what you are doing, interfere with what you are doing, and is noisy.

*Peer connection (PeC).* The average values of three questions on when and how respondents communicate with the friend via "phone conversation", "go over to one another residences", and "go together for a movie/skating/shopping/sports event" is evaluated as connection with friends. The answers are coded as "never", "once a month", "once per week", and "many times a week or everyday".

*Peer psychological control (PePC).* Psychological control of peers is quantified by the mean score of the answers to three questions on how often their peers disagree with you, how often they make you feel that your ideas are not as good as theirs, "humiliate or bug you" and "drag you down". The answers are coded as: "hardly ever", "once every month", "once per week", "a couple of times in a week", and "every day".

*Peer regulation (PeR).* Friend's regulation focuses on friends' influence in complying with laws and regulations. It is measured as the mean of two questions about how often the friend

"supports you do whatever is correct" and "motivates you to abide by the rules". "Never", "once a month", "once a week", "a few times a week and everyday" are the replies.

*Teacher's concerns (TC)*. The teacher's concern demonstrates teachers' supportive, friendly relationship with students that supports student's academic well-being. To measure teachers' concerns, four questions were asked about teachers' willingness to assist them with schoolwork and solve personal problems. The average of responses is used as a measure of the teachers' concerns.

*Student's academic performance (SAP)*. Since adolescents were interviewed from both household and school, it is difficult to assess academic performance forms the academic year performance. Therefore, students' academic success is assessed on an ordinary scale by a single question:' In general, how well are you doing in school? The replies are: 1 "way above average", 2 "a little above average", 3 "average", 4 "at just below average" and 5 "far below average".

**Reliability of dependent variable and contextual factors.** Reliability of contextual factors and knowledge on pubertal changes have been measured using Cronbach's alpha. The value of the Cronbach's α is 0.646 for knowledge on pubertal changes which is acceptable [37]. Among the contextual factors PBC (Cronbach's α = .70), CD (Cronbach's α = .75), CPC (Cronbach's α = .81), PeR (Cronbach's α = .79), PePC (Cronbach's α = .82), and TC (Cronbach's α = .88) had high reliability. Whereas, PLS (Cronbach's α = .63), and PeC (Cronbach's α = .61) had acceptable reliability.

## Statistical analysis

Simple descriptive analysis and multivariate statistical analysis were performed in this study. The categorical socio-demographic characteristics were presented by frequency and percentage distribution. The generalized structural equation model was also used to test the hypothetical relational path in which individual level, family level and community level variables correlated adolescents' knowledge on pubertal changes. There was no latent variable included in the model. Chi-square and t-test were used for the model specification purpose due to the scarcity of previous literature which considered the proximal process of adolescent's knowledge on puberty as the conceptual framework. Finally, diagonally weighted least squares (DWLS) had been used for estimating regression coefficients of structural equation model since DWLS estimation method provides a more accurate estimate for ordinal data.

All the analysis was conducted using the software R version 3.6.0. A Simultaneous equation model was fitted using the "lavaan" package [38].

## Results

### Socio-demographic characteristics of sampled adolescents

Table 1 demonstrates descriptive statistics of socio-demographic variables. There was almost an equal number of representatives of both sexes in the survey data (male 50.8% and female 49.2%). The mean age of adolescents and their years of schooling was 14.7 and 8.1,respectively. More than half of the adolescent's parents (father and mother) had achieved a secondary level education (51.2% for father and 57.3% for mother), whereas only 7.5 percent of adolescent's fathers had higher education and only a small portion of adolescent's mother completed higher education (2.7%).

### Contextual factors of adolescents

Descriptive statistics of contextual factors, Table 2, showed that parental behavioural control and community disorganization ranges from 1 to 3, and the remaining variables' values

**Table 1. Descriptive statistics of socio-demographic characteristics.**

| Variable | Group | Frequency/Mean±SD | Percentage |
|---|---|---|---|
| Year of Schooling | | 8.08±1.02 | |
| Age | | 14.67±1.32 | |
| Sex | Male | 1384 | 50.8 |
| | Female | 1340 | 49.2 |
| Religion | Islam | 2556 | 93.8 |
| | Others | 168 | 6.2 |
| Father's education | No education | 500 | 18.3 |
| | Primary | 623 | 22.9 |
| | Secondary | 1397 | 51.3 |
| | Higher | 204 | 7.5 |
| Mother's education | No education | 412 | 15.1 |
| | Primary | 677 | 24.9 |
| | Secondary | 1560 | 57.3 |
| | Higher | 75 | 2.7 |

ranging from 1 to 4 except parental limit setting which can take value from 1 to 4. Family level variables parental behavioural control and Parental limit setting had median values 2.67 and 2.75 respectively. The median of parental behavioural control was close to the maximum value indicates high parental behavioural control for the adolescents. The median value of peer psychological control was 1.25 which was very close to the lower limit of the range indicating adolescents who are merely psychologically controlled by the peer. But the peer regulation had median value 4 and for peer connection was 3.

## Findings from generalized structural equation model analysis

Model generating approach [39] had been used for the model specification. Initially based on Bronfenbrenner's bioecological model [30] all possible paths are included in our model. Then we went through a process of deleting and adding paths in the model based on the modification indices to arrive at a final best-fitting model that also fit well with substantive theoretical and practical meaning. Finally, the fitted structural equation model was over-identified as the model's degrees of freedom are 73, indicating that the number of free parameters to be estimated is less than the number of distinct values in the sample variance and covariance matrix. The model fit test statistic ($\chi^2$) was with p-value less than 0.001. The Goodness of fit index value (GFI = 0.89) and values of error-of-approximation based fit indices (RMSEA = 0.07, RMR = 0.02) indicated that the model was reasonably fit to the data.

**Table 2. Descriptive statistics of contextual factors.**

| Factors | Median | Minimum | Maximum |
|---|---|---|---|
| Student's academic performance | 0.67 | 1 | 5.00 |
| Parental behavioural control | 2.67 | 1 | 3.00 |
| Parental limit setting | 2.75 | 1 | 4.00 |
| Peer connection | 3.00 | 1 | 5.00 |
| Peer psychological control | 1.25 | 1 | 5.00 |
| Peer regulation | 4.00 | 1 | 5.00 |
| Community disorganization | 1.67 | 1 | 3.00 |
| Community psychological control | 2.67 | 1 | 5.00 |
| Teacher's concern | 4.00 | 1 | 5.00 |

**Table 3. Result of simultaneous equation model of contextual factors with knowledge about puberty.**

| Path | Estimate (Standardised) | P-value | Path | Estimate (Standardised) | P-value |
|---|---|---|---|---|---|
| KAP ←Female | 0.754 | <0.001 | EOS ←PLS | -0.053 | 0.011 |
| KAP ←Education | 0.578 | <0.001 | EOS ←PPC | -0.005 | 0.889 |
| KAP ←Age | 0.29 | <0.001 | SAP←ME1 | 0.139 | 0.001 |
| KAP ←Islam | -0.278 | 0.10 | SAP← ME2 | 0.108 | 0.006 |
| KAP ←SAP | -0.534 | 0.03 | SAP← ME3 | 0.094 | 0.010 |
| KAP ←PBC | 0.119 | 0.703 | SAP ←PBC | -0.054 | <0.001 |
| KAP ←PLS | -0.398 | <0.001 | SAP ←PLS | -0.030 | <0.001 |
| KAP ←PePC | -0.155 | 0.009 | SAP ←PPC | 0.006 | 0.592 |
| KAP ←PeC | -0.286 | <0.001 | PBC ←PeR | 0.073 | <0.001 |
| KAP ←PeR | 0.134 | 0.01 | PBC ←PePC | -0.064 | <0.001 |
| KAP ←TC | 0.069 | 0.102 | PBC ←CD | -0.064 | <0.001 |
| KAP ← ME1 | -0.872 | 0.04 | PBC ←TC | 0.074 | <0.001 |
| KAP ← ME2 | -0.483 | 0.200 | PLS ←PeC | 0.063 | <0.001 |
| KAP ← ME3 | -0.328 | 0.437 | PLS ←PeR | 0.007 | 0.603 |
| EOS ← ME1 | 0.078 | 0.518 | PLS ←CD | -0.048 | 0.052 |
| EOS ← ME2 | 0.047 | 0.137 | PLS ←CPC | -0.033 | 0.017 |
| EOS ← ME3 | 0.104 | 0.413 | PLS ←TC | 0.124 | <0.001 |
| EOS ←FWS | 0.016 | 0.755 | PPC ←PePC | 0.082 | <0.001 |
| EOS ←PBC | 0.016 | 0.634 | PPC ←CD | 0.10 | 0.561 |

KAP = Knowledge about Puberty; EOS = Year of Schooling; ME = Mother's Education; FE = Father's Education; SAP = Student' s Academic performance; PBC = Parental behavioural control; PLS = Parental limit setting; PePC = Peer psychological control; PeC = Peer connection; PeR = Peer regulation; TC = Teacher's concern.

The path coefficient of the structural equation model showed in Table 3. The result of structural equation model showed that gender (β = 0.754, P<0.001), education (β = 0.578, P<0.001), age (β = 0.29, P<0.001), and parental limit setting (β = 0.398, P<0.001) had significant effect on student's knowledge on pubertal changes.

There were several instances of mediation effect in the model. Student's academic performance act as a mediating variable for PPC→SAP→KAP, PLS→SAP→KAP, and PBC→SAP→KAP. Whereas, student's educational aspiration was the mediator for the indirect effect of the father's education (FE→SEA→KAP), mother's education (ME→SEA→KAP), and parental behavioural control (PBC→SEA→KAP) on knowledge about pubertal changes. The community-level variable, community disorganization showed an indirect impact on knowledge about pubertal changes through the mediating variable parental limit setting (CD→PLS→KAP). Adolescents' peer connection, regulation, and psychological control also demonstrated secondary effects on knowledge about pubertal changes through the mediating variables of parental limit settings and parental psychological control (PePC→PLS→KAP, PeR→PLS→KAP, and PePC→PBC→KAP).

## Effect of bio-social factors on knowledge on puberty (direct effects, indirect effects, and total effect)

Table 4 showed that sex, age, years of schooling and academic performances significantly influenced adolescents' knowledge of pubertal changes. The gender (β = 0.753, p<0.001) of adolescents directly correlated adolescents' understanding of puberty, which means that female adolescents have a higher understanding of puberty than their male counterparts. Adolescents'

**Table 4. Direct effect, indirect effect and total effect of bio-social factors on Knowledge on puberty.**

| Variable | Direct effect β (95% CI) | Indirect effect | | Total effect (95% CI) |
|---|---|---|---|---|
| | | β (95% CI) | Mediator Variable | |
| **Sex** | | | | |
| Female | 0.754*** [0.74, 0.768] | NA | NA | 0.754*** [0.74, 0.768] |
| Male | Ref. | Ref. | | Ref. |
| **Age** | 0.29*** [0.284, 0.296] | NA | NA | 0.29*** [0.284, 0.296] |
| **Religion** | | | | |
| Islam | -0.278 [-0.273, -0.283] | NA | NA | -0.278 [-0.273, -0.283] |
| Others | Ref. | Ref. | | Ref. |
| **Year of Schooling** | 0.578*** [0.567, 0.589] | NA | NA | 0.578*** [0.567, 0.589] |
| **Student' s Academic performance** | -0.534*** [-0.524, -0.544] | NA | NA | -0.534*** [-0.524, -0.544] |
| **Mother's education** | | | | |
| No education | -0.872* [-0.855, -0.889] | -0.076 [-0.075, -0.077] | Year of schooling, and Academic Performance | -0.949* [-0.931, -0.967] |
| Primary education | -0.483 [-0.474, -0.492] | -0.059 [-0.058, -0.06] | Year of schooling, and Academic Performance | -0.541 [-0.531, -0.551] |
| Secondary education | -0.328 [-0.322, -0.334] | -0.052 [-0.051, -0.053] | Year of schooling, and Academic Performance | -0.38 [-0.373, -0.387] |
| Higher education | Ref. | Ref. | | Ref. |
| **Parental behavioural control** | 0.119 [0.117, 0.121] | 0.029* [0.028, 0.03] | Academic Performance | 0.149 [0.146, 0.152] |
| **Parental limit setting** | -0.398*** [-0.39, -0.406] | -0.022 [-0.022, -0.022] | Year of schooling, and Academic Performance | -0.42*** [-0.412, -0.428] |
| **Peer connection** | -0.286*** [-0.28, -0.292] | -0.001 [-0.001, -0.001] | Parental limit setting | -0.287*** [-0.281, -0.293] |
| **Peer regulation** | 0.134* [0.131, 0.137] | 0.002 [0.002, 0.002] | Parental behavioural control | 0.134* [0.131, 0.137] |
| **Peer psychological control** | -0.155*** [-0.152, -0.158] | -0.002 [-0.002, -0.002] | Parental behavioural control | -0.156*** [-0.153, -0.159] |
| **Community psychological control** | NA | 0.001 [0.001, 0.001] | Parental limit setting | 0.001 [0.001, 0.001] |
| **Community disorganization** | NA | -0.001 [-0.001, -0.001] | Parental behavioural control and Parental limit setting | -0.001 [-0.001, -0.001] |
| **Teacher's concern** | 0.069*** [0.068, 0.07] | -0.001 [-0.001, -0.001] | Parental behavioural control and Parental limit setting | 0.068*** [0.067, 0.069] |

age (β = 0.293, p<0.001) had a significant positive impact on knowledge about pubertal changes, Table 4. The knowledge about pubertal changes increased by 0.293 for every year increase in the adolescents' age. Adolescents' years of schooling (β = 0.578, p<0.001) also showed a direct positive effect on their knowledge of pubertal changes. This means that as the adolescents spend one year more time at school, their knowledge of pubertal changes significantly increased by 0.578. However, students' academic performances (β = - 0.534, p<0.001) had a negative effect on knowledge about pubertal changes, which indicates that adolescents who were good at school had less knowledge on pubertal changes.

At the family level, the mother's education and parental limit-setting directly correlated adolescents' knowledge on pubertal changes. While parental behavioural control possessed an indirect effect on knowledge on pubertal changes. Mothers with no education had a direct (β = -0.872, p< 0.05) negative impact on adolescents' puberty knowledge than mothers with higher education. The analysis indicated that children with higher educated mothers had more pubertal knowledge. Similarly, parents' limitation setting on their child was directly (β = -0.398, p< 0.001), impeding the acquisition of the child's knowledge on pubertal changes. Unlike the

above two-variable parental behavioural control (β = 0.029, p< 0.05) indirectly enhanced adolescents' knowledge on pubertal changes.

Among the community level variables, peer connection, peer regulation and teacher's concern demonstrated a significant effect on adolescent's knowledge of pubertal changes. Peer psychological control, community disorganization, and community psychological control failed to significantly impact adolescents' knowledge of pubertal changes. Among the significant variables, peer connection (-0.286, p< 0.001) negatively influenced adolescent's knowledge of pubertal changes more strongly than the other two significant variables. However, peer regulation (β = 0.134, p< 0.05) and teacher's concern (0.069, p< 0.05) produces positive impact on adolescent's knowledge on pubertal changes.

## Discussion

The analysis of the study showed that adolescents' knowledge of pubertal changes is influenced significantly by various individual level, family level, and community-level factors. Sex, age, year of schooling, and academic performance were the individual level of factors that displayed a significant effect on adolescent's knowledge of pubertal changes. This current study revealed that knowledge of pubertal changes is positively influenced by age and years of schooling. A similar type of positive impact of age and year of schooling was also founded by Uddin and Choudhury [22]. As adolescents get older and complete more academic years, they are exposed more to the source of knowledge on pubertal changes and experience those pubertal changes as result they acquire higher knowledge on puberty with the increase in age and year of schooling. An interesting finding of our analysis showed that female adolescents had more knowledge on puberty than males. This evidence is supported by an earlier study conducted in Bangladesh and the study observed that male adolescents had poor knowledge of pubertal changes than female adolescents [40]. Separate studies from India on school-going adolescents [41] and Portugal on college students [42] found the same findings. The higher presence of consciousness regarding the pubertal changes among female adolescents may be related to the early onset of puberty among female adolescents, extensive focus to the female adolescents of adolescents sexual and reproductive health awareness-raising program, and the variation in the main source from which they get the information regarding the pubertal changes.

This study revealed that religion plays an important role in the knowledge of pubertal changes. Muslim students had lower pubertal change knowledge than students of other religions. This dissimilarity between different religions may be due to the different views towards human sexuality among different religions. In socio-economic context in south Asian countries, discussion about human sexuality is often recognized as taboo and the socio-religious context refrains adults from discussing with adolescents [43]. A previous study found that Hindu students had 1.76 times more likely to have better communication with mothers regarding sexual and reproductive health (SRH) [1].

Among family-level factors, Mother's education shows a positive impact on adolescent's pubertal knowledge. These findings are also consistent with the study of Uddin and Choudhury [21] on adolescent girls in rural areas in Bangladesh and another study in India [15]. This is because daughters also have a trustworthy relationship with their mother as a primary source of SRH information [15]. Though mothers are uncomfortable discussing SRH-related problems in our social structure, educated mothers do not hesitate to talk about SRH issues [44]. However, parental limit setting on children significantly reduces adolescents' pubertal knowledge in this analysis. This finding depicts our country's tradition, where parents think that pubertal changes are a natural human development phenomenon that needs to remain secret [45]. Parental behavioural control showed an indirect positive effect on adolescent's

knowledge on pubertal changes. This indicates that parents by imposing restrictions on adolescents' behaviour increase their knowledge of pubertal changes through the mediating variable of years of schooling and academic performance.

Peers and community are two important social components that have a great impact on adolescent functioning. This analysis found that despite being a major source of knowledge regarding puberty, peer connection significantly reduces adolescents' pubertal knowledge. It might have occurred due to incomplete and misleading information. The findings of Das and Roy [17]- peer almost always supplies incomplete and fantasy-oriented sexual and reproductive knowledge, also supports the findings of this study regarding the role of peer connection. On the other hand, peer regulation showed a direct positive impact on adolescents' pubertal knowledge.

Adolescents spend a significant portion of their daily life in school. Therefore, school-level factors are essential for shaping their knowledge. Among other school-level factors, students' academic performance negatively impacts adolescent's knowledge of puberty. This contrary result can be due to the ignorance of students to the topics related to reproductive health issues as there was no exam taken on physical education and health in Bangladesh. On the other hand, the teacher's concern demonstrates an overall positive impact on adolescent's pubertal knowledge. Furthermore, the findings demonstrated that teacher concern had a significant positive impact on adolescent's pubertal knowledge. This evidence coincided with other studies and observed that when teachers exhibit care towards students, the adolescents are motivated to learn and eliminate and correct disruptive behaviour [46, 47].

## Conclusions

In summary, we could conclude that sex, years of schooling, age, parental limit-setting, peer concern, and peer regulation were the significant determinants on adolescent's knowledge of pubertal changes. Male and Muslim adolescents had less understanding regarding the pubertal changes. Whereas, academic performance also demonstrated a negative relationship with knowledge on puberty. Though, the social context parents and peers were negatively associated with adolescent's knowledge on puberty, Teachers are positively associated with knowledge on puberty. This research establishes a new framework for policy research to develop appropriate strategies to improve SRH conditions through cultural relevance. This study also recommended that policymakers and development strategies need to formulate appropriate strategies that can change parents and peers' roles to ensure adolescent's informed puberty. Furthermore, there is a need to evaluate the effectiveness of the awareness raising program considering the conceptual framework used in this study.

## Limitations

There are several limitations in this study. The study sample was taken from only rural schools which limits the generalization of the findings. The academic performance was assessed using participants self-response. Adolescents' responses may vary by school, but any random effect of the school had not been included in this study.

## Acknowledgments

The authors are gratefully thankful to the Bangladesh Bureau of Educational Information and Statistics (BANBEIS), Ministry of Education, Bangladesh, and Department of Statistics of Jagannath University.

## Author Contributions

**Conceptualization:** Md Injamul Haq Methun, M. Sheikh Giash Uddin.

**Data curation:** M. Sheikh Giash Uddin.

**Formal analysis:** Md Injamul Haq Methun, Md. Ismail Hossain, Md. Jakaria Habib.

**Methodology:** M. Sheikh Giash Uddin.

**Writing – original draft:** Md Injamul Haq Methun, Ahmed Abdus Saleh Saleheen, Iqramul Haq.

**Writing – review & editing:** Md Injamul Haq Methun, Md. Ismail Hossain, Iqramul Haq.

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
