## [Decision Letter · Decision Letter 0]

24 Jun 2021

PONE-D-21-15883

Impact of School-based Sexual and Reproductive Health Education among Secondary Student’s Knowledge on Pubertal Changes in Rural Bangladesh

PLOS ONE

Dear Dr. Haq,

Thank you for submitting your manuscript to PLOS ONE. After careful consideration, we feel that it has merit but does not fully meet PLOS ONE’s publication criteria as it currently stands. Therefore, we invite you to submit a revised version of the manuscript that addresses the points raised during the review process.

The study raised important and sensitive topic; focusing on the ``Impact of School-based Sexual and Reproductive Health Education among Secondary Student’s Knowledge on Pubertal Changes in Rural Bangladesh``. However, fundamental issues are indicated to be provided mainly for the methodology section and the statistical analysis.  The manuscript could be greatly strengthened by aggressive editing for most of the study sections. Consider revising the spelling and grammar throughout the manuscript for increased clarity. 

Please note that your manuscript was reviewed by 4 experts in the field. There is consensus agreement that the idea of the article is interesting but also the majority detected sections that required additional work. The reviewers identified many important problems and provided copious comments (enclosed).

The manuscript could be greatly strengthened by considering editing according to the specific Reviewers’ comments.

Please submit your revised manuscript by July 30, 2021, 11:59 PM. If you will need more time than this to complete your revisions, please reply to this message or contact the journal office at plosone@plos.org. Please include the following items when submitting your revised manuscript:

We look forward to receiving your revised manuscript.

Kind regards,

Ammal Mokhtar Metwally, Ph.D (MD)

Academic Editor

PLOS ONE

Journal Requirements:

4. Please ensure that you refer to Figure 1 in your text as, if accepted, production will need this reference to link the reader to the figure.

5. Please include a copy of Table 5 which you refer to in your text on page 11.

Reviewers' comments:

Reviewer's Responses to Questions

**Comments to the Author**

1. Is the manuscript technically sound, and do the data support the conclusions?

Reviewer #1: Yes

Reviewer #2: Yes

Reviewer #3: Partly

Reviewer #4: Partly

2. Has the statistical analysis been performed appropriately and rigorously? 

Reviewer #1: Yes

Reviewer #2: Yes

Reviewer #3: I Don't Know

Reviewer #4: Yes

3. Have the authors made all data underlying the findings in their manuscript fully available?

Reviewer #1: No

Reviewer #2: Yes

Reviewer #3: No

Reviewer #4: No

4. Is the manuscript presented in an intelligible fashion and written in standard English?

Reviewer #1: Yes

Reviewer #2: Yes

Reviewer #3: No

Reviewer #4: Yes

5. Review Comments to the Author

Reviewer #1: The manuscript is technically sound and the statistical analysis has been performed appropriately but they have mentioned that data cannot be shared publicly because of the restriction of data authority. However, they can add data course link. The manuscript needs major revisions as sentences are not very clear.

Reviewer #2: The authors have touched a very sensitive issue related to Adolescent in a school setting. Few of the comments below can be considered to make the paper more useful for the readers and other users.

1. The justification given as a rational for the study- "No specific research has been done so far on how parents and our society hinder adolescent's knowledge of pubertal changes."- seems a bit odd. There are several published studies done in rural Bangladesh and several Asian countries and probably specific research might not be done in the specific study area in the past, and it should not be generalized.

2. I am not clear if there was a parental consent since the mean age in this study was 14.67 Years. Did the researchers used other mechanisms to ensure parental consent?

3. One of the variables relate to Student's academic performance by directly asking the student. It might have been better if the actual performance form the academic year was assessed.

4. The discussion tried to compare results with other studies but the why part of the results are not included in the discussion part. As an example the female adolescent's understanding of puberty was higher that male, which might be related to having the onset menses (Menstrual period) during that age under study. So such justifications are important to give the reader in knowing what reasons are linked to the findings of the study.

5. I don't see any limitation of the study included to justify some of the drawbacks which might be very useful for the reader to understand those factors which hinder the researchers.

6. Part of the recommendation can include possible future studies to be done based on the findings of the study.

7. The title of the study relates to impact of school based SRH education, but this is not clearly seen in the study and it might be good to modify the title to represent the study methodology and findings.

Reviewer #3: The paper "Impact of School-based Sexual and Reproductive Health Education among Secondary Student’s Knowledge on Pubertal Changes in Rural Bangladesh" undertakes an interesting topic. It seems like the study has certain potential. The results are interesting and it seems that authors were able to put their findings within the general literature context. However, the study lacks considerably in terms of readability and academic rigor of presentation. Below I present my more specific remarks but I encourage you to thoroughly proof read the manuscript.

Abstract

"parental limit setting" should be explained more thoroughly.

Please refrain from using causal language (e.g., influencing) in your cross-sectional study.

Your claim that some of your variables were "influencing" knowledge on puberty both directly and indirectly suggests that you were using some mediation analysis. The article would benefit if you were more explicit about that fact and also, if you could state the reasons for running mediation analysis.

Instead of simple conclusions, you could provide some policy implications to improve knowledge about pubertal changes in Bangladesh.

Introduction

What do you mean by "To make healthy adults from childhood, ..."? This sentence is very unclear. I would suggest to omit first three sentences in your introduction.

You claim that "society is responsible for ensuring informed puberty" but probably even more responsibility lies in the family and in the schooling system.

Please refrain from using "condition" in relation to puberty. It is rather a developmental stage.

Again, in stating your objectives please refrain from causal language (e.g., affect).

Conceptual Framework

It would be worth describing which elements of bioecological model of Bronfenbrenner were included in your framework and what is the conceptual novelty of your approach. In the current version it is not clear.

The section on data is not clear. Individuals between 10 and 18 or 10 and 24 were sampled? 3014 or 3000 were surveyed? 1504+1509 is not equal to 3014. "VI to IX" or "6-9"?

It is not clear while "outcome variable" is under the heading of "study population". Please explain.

Knowledge of pubertal changes scale should be somehow described. Questionnaire should be put in the appendix. If the scale is used, its psychometric properties should be checked - at least reliability of the scale.

The scales used in the study (PBC, PLS, CD, CPC, PeC, PePC, PeR, TC) should have their psychometric properties checked. Reader should also have access to detailed wording of the items comprising each scale.

Statistical Analysis

Again, please refrain from causal language.

Please elaborate more on specification and estimation of your structural equation model.

Why you have not decided to use latent constructs in your model? Your data is perfectly fit to such an approach.

Results

In the analysis of contextual factors why do you use median when for the socio-economic characteristics you were using mean?

Table 3 contains variable KAP, it is not clear what it means?

Lack of reference categories for variables like "Gender", "Religion" precludes understanding of the results in Table 3.

When presenting indirect effects in Table 4, I would suggest to include mediating variable. It is difficult to understand to which indirect paths the results refer.

Discussion

What is the meaning of SRH? The acronym has not been introduced.

Reviewer #4: Question 1: The Sexual and Reproductive Health Education variable was not included in the questionnaire and in data analysis. The conclusion is based on this intervention variable which was not part of the analysis

Question 3: Some restriction applied on data availability

6. PLOS authors have the option to publish the peer review history of their article (what does this mean?). If published, this will include your full peer review and any attached files.

Reviewer #1: **Yes: **Nikita Bhattarai

Reviewer #2: **Yes: **Mengistu Asnake Kibret

Reviewer #3: No

Reviewer #4: **Yes: **Tekachew Wana

---

## [Author Response · Author response to Decision Letter 0]

30 Jul 2021

Response to the editors remarks:

Response: We have followed the instruction of PLOS ONE style

Response: In method section we have clearly stated the consent which are taken before data collections.

[line 162 -163]

Response: ORCID id have been attached.

4. Please ensure that you refer to Figure 1 in your text as, if accepted, production will need this reference to link the reader to the figure.

5. Please include a copy of Table 5 which you refer to in your text on page 11.

Response: There was mistake, actually it was Table -4. We have amended the Table number.

Responses to the reviewers’ remarks: 

The manuscript is technically sound and the statistical analysis has been performed appropriately but they have mentioned that data cannot be shared publicly because of the restriction of data authority. However, they can add data course link. The manuscript needs major revisions as sentences are not very clear.

Response: We revised the manuscript. We revised the manuscript. 

Reviewer-2

1. The justification given as a rational for the study- "No specific research has been done so far on how parents and our society hinder adolescent's knowledge of pubertal changes."- seems a bit odd. There are several published studies done in rural Bangladesh and several Asian countries and probably specific research might not be done in the specific study area in the past, and it should not be generalized.

Response: We have amended the justification. [Line: 104-107] We have amended the justification. [Line: 104-107]

2. I am not clear if there was a parental consent since the mean age in this study was 14.67 Years. Did the researchers used other mechanisms to ensure parental consent?

Response: Parental consent was taken. We have made an attempt to clarify this issue in line 161-163. Parental consent was taken. We have made an attempt to clarify this issue in line 161-163. 

3. One of the variables relate to Student's academic performance by directly asking the student. It might have been better if the actual performance forms the academic year was assessed.

Response: Since adolescents were interviewed from both household and school, it is difficult to assess academic performance forms the academic year performance. Therefore, Student's academic performance by directly asking the student. [Line: 234 – 237] Since adolescents were interviewed from both household and school, it is difficult to assess academic performance forms the academic year performance. Therefore, Student's academic performance by directly asking the student. [Line: 234 – 237]

4. The discussion tried to compare results with other studies but the why part of the results are not included in the discussion part. As an example, the female adolescent's understanding of puberty was higher that male, which might be related to having the onset menses (Menstrual period) during that age under study. So such justifications are important to give the reader in knowing what reasons are linked to the findings of the study.

Response: We have revised the discussion section of the manuscript. We have revised the discussion section of the manuscript. 

5. I don't see any limitation of the study included to justify some of the drawbacks which might be very useful for the reader to understand those factors which hinder the researchers.

Response: We have added limitations of the study. 

 We have added limitations of the study. 

6. Part of the recommendation can include possible future studies to be done based on the findings of the study.

Response: We have revised the recommendation section. We have revised the recommendation section. 

7. The title of the study relates to impact of school based SRH education, but this is not clearly seen in the study and it might be good to modify the title to represent the study methodology and findings.

Response: We have modified the title as “Biosocial correlates of adolescent’s knowledge on pubertal changes in rural Bangladesh: a structural Equation model”

 We have modified the title as “Biosocial correlates of adolescent’s knowledge on pubertal changes in rural Bangladesh: a structural Equation model”

Reviewer-3

1. The paper "Impact of School-based Sexual and Reproductive Health Education among Secondary Student’s Knowledge on Pubertal Changes in Rural Bangladesh" undertakes an interesting topic. It seems like the study has certain potential. The results are interesting and it seems that authors were able to put their findings within the general literature context. However, the study lacks considerably in terms of readability and academic rigor of presentation. Below I present my more specific remarks but I encourage you to thoroughly proof read the manuscript.

Response: We have revised our manuscript to improve the readability of the manuscript. We have revised our manuscript to improve the readability of the manuscript. 

We have replaced "parental limit setting" with “parental limit setting on daily activities ” in the result of the abstract. [ line 22-23]. Moreover, the explanation of “parental limit setting is given in contextual factors under methodology [line: 197 to 203] We have replaced "parental limit setting" with “parental limit setting on daily activities ” in the result of the abstract. [ line 22-23]. Moreover, the explanation of “parental limit setting is given in contextual factors under methodology [line: 197 to 203]

2. Please refrain from using causal language (e.g., influencing) in your cross-sectional study.

Response: We have revised our manuscript and replace the causal language. [line no. 53,55] We have revised our manuscript and replace the causal language. [line no. 53,55]

3. Your claim that some of your variables were "influencing" knowledge on puberty both directly and indirectly suggests that you were using some mediation analysis. The article would benefit if you were more explicit about that fact and also, if you could state the reasons for running mediation analysis.

Response: We have included the reasons of running mediation analysis in the methodology section of the abstract [line 40 to 43] We have included the reasons of running mediation analysis in the methodology section of the abstract [line 40 to 43]

4. Instead of simple conclusions, you could provide some policy implications to improve knowledge about pubertal changes in Bangladesh.

Response: We have revised the conclusion section of the manuscript We have revised the conclusion section of the manuscript. 

Introduction

5. What do you mean by "To make healthy dults from childhood, ..."? This sentence is very unclear. I would suggest to omit first three sentences in your introduction.

Response: We have omitted the first three line of the introduction. We have omitted the first three line of the introduction. 

6. You claim that "society is responsible for ensuring informed puberty" but probably even more responsibility lies in the family and in the schooling system.

Response: Replacing Society with family. 

[line: 87] Replacing Society with family. 

[line: 87]

7. Please refrain from using "condition" in relation to puberty. It is rather a developmental stage.

Response: We have made an amendment to refrain from using causal language. We have made an amendment to refrain from using causal language. 

8. Again, in stating your objectives please refrain from causal language (e.g., affect).

Response: We have made an amendment to refrain from using causal language We have made an amendment to refrain from using causal language

10. Conceptual Framework

It would be worth describing which elements of bioecological model of Bronfenbrenner were included in your framework and what is the conceptual novelty of your approach. In the current version it is not clear.

Response: We have elaborately described the Conceptual framework section. We have elaborately described the Conceptual framework section.

11.The section on data is not clear. Individuals between 10 and 18 or 10 and 24 were sampled? 3014 or 3000 were surveyed? 1504+1509 is not equal to 3014. "VI to IX" or "6-9"?

Response: Individual between 10-24 were sampled. 3013 were surveyed. We have revised our manuscript with VI-IX. Individual between 10-24 were sampled. 3013 were surveyed. We have revised our manuscript with VI-IX.

12. It is not clear while "outcome variable" is under the heading of "study population". Please explain.

Response: The manuscript is rewritten to increase the clarity of the manuscript. The manuscript is rewritten to increase the clarity of the manuscript.

13. Knowledge of pubertal changes scale should be somehow described. Questionnaire should be put in the appendix. If the scale is used, its psychometric properties should be checked - at least reliability of the scale.

Response: We included the reliability. [ line: 241-247] We included the reliability. [ line: 241-247]

14. The scales used in the study (PBC, PLS, CD, CPC, PeC, PePC, PeR, TC) should have their psychometric properties checked. Reader should also have access to detailed wording of the items comprising each scale.

Response: We included the reliability. [ line: 241-247] We included the reliability. [ line: 241-247]

Statistical Analysis

15. Again, please refrain from causal language.

Response: We have made an amendment to refrain from using causal language. We have made an amendment to refrain from using causal language.

16. Please elaborate more on specification and estimation of your structural equation model.

Response: The statistical analysis section have been rewritten with an elaboration of specification and estimation of structural equation model. [ Line: 256-261] The statistical analysis section have been rewritten with an elaboration of specification and estimation of structural equation model. [ Line: 256-261]

17. Why you have not decided to use latent constructs in your model? Your data is perfectly fit to such an approach.

Response: We have included the explanation of not using latent construct between line 176 to 183. We have included the explanation of not using latent construct between line 176 to 183. 

Results

18. In the analysis of contextual factors why do you use median when for the socio-economic characteristics you were using mean?

Response: For continuous socio-economic characteristics, mean have been used and for ordinal contextual factors median have been used. We have included the explanation between line 2513 to 253. For continuous socio-economic characteristics, mean have been used and for ordinal contextual factors median have been used. We have included the explanation between line 2513 to 253.

19.Table 3 contains variable KAP, it is not clear what it means?

Response: We made an attempt to clarify the meaning the variable We made an attempt to clarify the meaning the variable

20. Lack of reference categories for variables like "Gender", "Religion" precludes understanding of the results in Table 3.

Response: We made an amendment of Table 3 with the reference categories. We made an amendment of Table 3 with the reference categories. 

21. When presenting indirect effects in Table 4, I would suggest to include mediating variable. It is difficult to understand to which indirect paths the results refer.

Response: We have included a column of mediator variables in Table 4. We have included a column of mediator variables in Table 4.

22. Discussion

What is the meaning of SRH? The acronym has not been introduced.

Response: We have amended the discussion section. [line 374] We have amended the discussion section. [line 374]

Reviewer-4

Question 1: The Sexual and Reproductive Health Education variable was not included in the questionnaire and in data analysis. The conclusion is based on this intervention variable which was not part of the analysis

Response: We have revised our manuscript. We have revised our manuscript. 

Question 3: Some restriction applied on data availability

---

## [Decision Letter · Decision Letter 1]

3 Sep 2021

PONE-D-21-15883R1

Biosocial correlates of adolescent’s knowledge on pubertal changes in rural Bangladesh: a structural Equation model

PLOS ONE

Dear Dr. Methun,

Thank you for submitting your manuscript to PLOS ONE. After careful consideration, we feel that it has merit but does not fully meet PLOS ONE’s publication criteria as it currently stands. Therefore, we invite you to submit a revised version of the manuscript that addresses the points raised during the review process.

Great effort was made by the authors to utilize the feedback that was provided for them to correct. I find it interesting and improved with respect to the original submission. However, there are few minor things to adjust and one major.  Discussion about the specification of the structural equation model (which is vital) is still missing. Please elaborate on your understanding on structural relationships, especially for the part of the model where you.

Please consider reviewers’ comments for more details   

We look forward to receiving your revised manuscript.

Kind regards,

Ammal Mokhtar Metwally, Ph.D (MD)

Academic Editor

PLOS ONE

Reviewers' comments:

Reviewer's Responses to Questions

**Comments to the Author**

1. If the authors have adequately addressed your comments raised in a previous round of review and you feel that this manuscript is now acceptable for publication, you may indicate that here to bypass the “Comments to the Author” section, enter your conflict of interest statement in the “Confidential to Editor” section, and submit your "Accept" recommendation.

Reviewer #1: All comments have been addressed

Reviewer #2: All comments have been addressed

Reviewer #3: (No Response)

2. Is the manuscript technically sound, and do the data support the conclusions?

Reviewer #1: Yes

Reviewer #2: Yes

Reviewer #3: Partly

3. Has the statistical analysis been performed appropriately and rigorously? 

Reviewer #1: Yes

Reviewer #2: Yes

Reviewer #3: Yes

4. Have the authors made all data underlying the findings in their manuscript fully available?

Reviewer #1: Yes

Reviewer #2: Yes

Reviewer #3: (No Response)

5. Is the manuscript presented in an intelligible fashion and written in standard English?

Reviewer #1: Yes

Reviewer #2: Yes

Reviewer #3: Yes

6. Review Comments to the Author

Reviewer #1: All the comments have been addressed, the datas were all available and it has been corrected. The manuscript had been written in standard English.

Reviewer #2: The authors have addressed all the comments and queries provided from my side and don't have additional comments.

Reviewer #3: At first I would like to thank the Authors for addressing the comments raised during the first stage of review. I have a number of minor remarks and one major concern. As for my major reservation, I am still hesitant about the structural model. Authors do not present the initial specification of the structural model and additionally they do not properly justify the links between variables. It is especially concerning for the mediation effects, which should be thoroughly discussed before being implemented in the analysis.

I would suggest revising the statements in lines 175-177 "Latent construct approach is the most suitable approach for measuring. However, to construct a latent variable it is essential to remove observed variables if the correlations between variables is very high or is not high-enough." It is not clear what is the intention behind the statement.

Could you provide a bit more elaborate description of the Kalule-Sabiti method. 2-3 additional sentences could help better understand the approach.

In line 287 you mention that "The structural equation model was over-identified," but it is unclear what kind of specification you suggest for the model. I would suggest that you draw your model with all potential covariates to show your starting specification.

Minor:

"pubertal period" sounds a bit artificial - please change this.

Line 84: should be "prevent"

Line 90 "think" sounds a bit strange in this context.

Line 91: Maybe instead of "mass media", current teenagers refer more to social media.

Line 101: Replace "find out" with "identify"

Line 130: Replace "carried out" with "provided"

Line 139: Revise the sentence.

Figure 1: "framework" instead of "frame work"

Line 222-223: The inverted commas system is not clear.

Line 242: "using" instead of "sing"

Line 163: Shouldn't it be "approved by" instead of "taken from"?

7. PLOS authors have the option to publish the peer review history of their article (what does this mean?). If published, this will include your full peer review and any attached files.

Reviewer #1: **Yes: **Nikita Bhattarai

Reviewer #2: **Yes: **Mengistu Asnake Kibret

Reviewer #3: No

---

## [Author Response · Author response to Decision Letter 1]

6 Nov 2021

Responses to the remarks of reviewers: 

1.I would suggest revising the statements in lines 175-177 "Latent construct approach is the most suitable approach for measuring. However, to construct a latent variable it is essential to remove observed variables if the correlations between variables is very high or is not high-enough." It is not clear what is the intention behind the statement.

Response: We have revised the lines 175-177. 

2.Could you provide a bit more elaborate description of the Kalule-Sabiti method. 2-3 additional sentences could help better understand the approach.

Response: We have a brief description of Kalule-Sabiti method in lines 191 to 196. 

3. In line 287 you mention that "The structural equation model was over-identified," but it is unclear what kind of specification you suggest for the model. I would suggest that you draw your model with all potential covariates to show your starting specification.

Response: We have explained the specification procedure in lines 293 – 297. 

Minor:

"pubertal period" sounds a bit artificial - please change this.

Response: In line 78 and 62, we replace “pubertal period” with “puberty”. 

In line 78 changes with puberty. In line 62 changes with puberty

Line 84: should be "prevent"

Response: Replaced. 

Line 90 "think" sounds a bit strange in this context.

Response: In line 90, we replace “think” with “Parents left”. 

Line 91: Maybe instead of "mass media", current teenagers refer more to social media.

Response: Yes, we have amended the line. 

Line 101: Replace "find out" with "identify"

Response: Replaced with identify

Line 130: Replace "carried out" with "provided"

Response: Replaced with “provided”. 

Line 139: Revise the sentence.

Figure 1: "framework" instead of "frame work"

Response: Replaced with “framework”

Line 222-223: The inverted commas system is not clear.

Response: The inverted commas system is rearranged.

Line 242: "using" instead of "sing"

Response: Replaced with “using”.

Line 163: Shouldn't it be "approved by" instead of "taken from"?

Response: Replaced with “approved by”.

---

## [Decision Letter · Decision Letter 2]

9 Dec 2021

PONE-D-21-15883R2Biosocial correlates of adolescent’s knowledge on pubertal changes in rural Bangladesh: a structural Equation modelPLOS ONE

Dear Dr. Methun,

Thank you for submitting your manuscript to PLOS ONE. After careful consideration, we feel that it has merit but does not fully meet PLOS ONE’s publication criteria as it currently stands. Therefore, we invite you to submit a revised version of the manuscript that addresses the points raised during the review process.

Please note that further language improvement is highly indicated in the manuscript to be considered as a sound one. Consider revising the spelling, grammar, diction, and syntax throughout the manuscript for increased clarity to meet the standards for PLOS one publication. 

We look forward to receiving your revised manuscript.

Kind regards,

Ammal Mokhtar Metwally, Ph.D (MD)

Academic Editor

PLOS ONE

Reviewers' comments:

Reviewer's Responses to Questions

**Comments to the Author**

1. If the authors have adequately addressed your comments raised in a previous round of review and you feel that this manuscript is now acceptable for publication, you may indicate that here to bypass the “Comments to the Author” section, enter your conflict of interest statement in the “Confidential to Editor” section, and submit your "Accept" recommendation.

Reviewer #1: All comments have been addressed

Reviewer #3: (No Response)

2. Is the manuscript technically sound, and do the data support the conclusions?

Reviewer #1: Yes

Reviewer #3: Yes

3. Has the statistical analysis been performed appropriately and rigorously? 

Reviewer #1: Yes

Reviewer #3: Yes

4. Have the authors made all data underlying the findings in their manuscript fully available?

Reviewer #1: Yes

Reviewer #3: (No Response)

5. Is the manuscript presented in an intelligible fashion and written in standard English?

Reviewer #1: Yes

Reviewer #3: Yes

6. Review Comments to the Author

Reviewer #1: All the comments have been addressed and the manuscript has been written in an intelligible fashion.

Reviewer #3: Dear Authors,

Thank you for your work. I still feel that there are some issues (even if small) that should be addressed before the article is published.

Please be more specific in your statement "parents and peers are negatively associated with adolescent’s understanding of pubertal changes". It is not clear what you mean.

Line 68 "in girls" seems redundant.

Line 77 - "recognize" is not a correct word here (maybe "accept" or "endorse" would be better)

Line 82 - "expose" needs a pronoun here

Line 84 - "prevents" is not correct

Lines 92-94 - the sentences are not clear

"upzila"'s meaning should be explained

Line 182 - still not clear what you mean when you say "using above approach"

Line 186 - instead of saying "independent variables" I would suggest "control variables"

Please read carefully your description of "Contextual factors". It requires a through proof reading.

Line 281 - what do you mean by saying "parental limit setting ranging from 1 to 4 and the upper limit for the remaining variables is 5"?

Line 300 - what is the "model fit test statistic" that you are referring to?

The values of model fit (GFI, RMSEA) are rather acceptable and not good.

In notes for Table 3, please describe the abbreviations used.

Table 3 - please comment whether the coefficients are standardized or unstandardized.

Please avoid causal language in your analysis. Your sample is cross-sectional and words like "affected" should not be used. Replace with "correlated", for example. Check the whole manuscript and eliminate causal language.

I see discrepancies between the coefficients presented in Table 3 and in the description provided in "Direct effects, indirect effects, and total effect". Please correct it.

7. PLOS authors have the option to publish the peer review history of their article (what does this mean?). If published, this will include your full peer review and any attached files.

Reviewer #1: **Yes: **Nikita Bhattarai

Reviewer #3: No

---

## [Author Response · Author response to Decision Letter 2]

24 Jan 2022

Thank you for giving us the opportunity to submit a revised draft of my manuscript titled “Biosocial correlates of adolescent’s knowledge on pubertal changes in rural Bangladesh: a structural Equation model” to Plos One. We appreciate the time and effort that you and the reviewers have dedicated to providing your valuable feedback on my manuscript. We are grateful to the reviewers for their insightful comments on my paper. We have been able to incorporate changes to reflect most of the suggestions provided by the reviewers. Here is a point-by-point response to the reviewers’ comments and concerns.

Responses to the remarks of reviewers: 

1.Please be more specific in your statement "parents and peers are negatively associated with adolescent’s understanding of pubertal changes". It is not clear what you mean.

Response: We have revised the line. (Line 50 and 51)

2. Line 68 "in girls" seems redundant.

Response: line 60: we deleted "in girls".

3. Line 77 - "recognize" is not a correct word here (maybe "accept" or "endorse" would be better)

Response: line 69 – “recognize’ replaced with “accept”.

4. Line 82 - "expose" needs a pronoun here

Response: line 74- We have added pronoun.

5. Line 84 - "prevents" is not correct

Response: line 76 – we replaced “prevents”.

6. Lines 92-94 - the sentences are not clear

Response: line 84 – 88 – we have amended the sentences. 

7. "upzila"'s meaning should be explained

Response: We added the meaning of “upazila” in line 135.

8. Line 182 - still not clear what you mean when you say "using above approach"

Response: line 176-177 – We have revised the sentence. 

9. Line 186 - instead of saying "independent variables" I would suggest "control variables"

Response: line 179 - "independent variables" replaced with "control variables".

10. Please read carefully your description of "Contextual factors". It requires a through proof reading.

Response: We have revised contextual factors.

11. Line 281 - what do you mean by saying "parental limit setting ranging from 1 to 4 and the upper limit for the remaining variables is 5"?

Response: The line is revised. (Line 277-278.)

12. Line 300 - what is the "model fit test statistic" that you are referring to?

Response: We have included in line 296.

13. The values of model fit (GFI, RMSEA) are rather acceptable and not good.

Response: We have made amendment the statement. (Line 297 -299).

14. In notes for Table 3, please describe the abbreviations used.

Response: We have added the abbreviations.

15. Table 3 - please comment whether the coefficients are standardized or unstandardized.

Response: standardized

16. Please avoid causal language in your analysis. Your sample is cross-sectional and words like "affected" should not be used. Replace with "correlated", for example. Check the whole manuscript and eliminate causal language.

Response: we made an amendment

17. I see discrepancies between the coefficients presented in Table 3 and in the description provided in "Direct effects, indirect effects, and total effect". Please correct it.

Response: we revised the manuscript.

---

## [Decision Letter · Decision Letter 3]

14 Feb 2022

Biosocial correlates of adolescent’s knowledge on pubertal changes in rural Bangladesh: a structural Equation model

PONE-D-21-15883R3

Dear Dr. Methun,

We’re pleased to inform you that your manuscript has been judged scientifically suitable for publication and will be formally accepted for publication once it meets all outstanding technical requirements.

Kind regards,

Ammal Mokhtar Metwally, Ph.D (MD)

Academic Editor

PLOS ONE

Additional Editor Comments (optional):

Reviewers' comments:

Reviewer's Responses to Questions

**Comments to the Author**

1. If the authors have adequately addressed your comments raised in a previous round of review and you feel that this manuscript is now acceptable for publication, you may indicate that here to bypass the “Comments to the Author” section, enter your conflict of interest statement in the “Confidential to Editor” section, and submit your "Accept" recommendation.

Reviewer #3: All comments have been addressed

2. Is the manuscript technically sound, and do the data support the conclusions?

Reviewer #3: Yes

3. Has the statistical analysis been performed appropriately and rigorously? 

Reviewer #3: Yes

4. Have the authors made all data underlying the findings in their manuscript fully available?

Reviewer #3: (No Response)

5. Is the manuscript presented in an intelligible fashion and written in standard English?

Reviewer #3: Yes

6. Review Comments to the Author

Reviewer #3: Thank you for your hard work. My only concern regards the figure 1. In its current form, it suggests that indirect path can run from "community level factors" to "mother's education", which seems pretty unlikely and the causal path is probably not set in a right direction. I would also suggest a proof editing before the final publication. In line 277 something is missing.

7. PLOS authors have the option to publish the peer review history of their article (what does this mean?). If published, this will include your full peer review and any attached files.

Reviewer #3: No

---

## [Editor Report · Acceptance letter]

9 Mar 2022

PONE-D-21-15883R3 

Biosocial correlates of adolescent’s knowledge on pubertal changes in rural Bangladesh: a structural Equation model 

Dear Dr. Methun:

I'm pleased to inform you that your manuscript has been deemed suitable for publication in PLOS ONE. Congratulations! Your manuscript is now with our production department. 

Kind regards, 

on behalf of

Professor Ammal Mokhtar Metwally 

Academic Editor

PLOS ONE